# Impact of Postharvest Putrescine Treatments on Phenolic Compounds, Antioxidant Capacity, Organic Acid Contents and Some Quality Characteristics of Fresh Fig Fruits during Cold Storage

**DOI:** 10.3390/plants12061291

**Published:** 2023-03-13

**Authors:** Emine Kucuker, Erdal Aglar, Mustafa Sakaldaş, Fatih Şen, Muttalip Gundogdu

**Affiliations:** 1Agriculture Faculty Department of Horticulture, Siirt University, Siirt 56100, Turkey; 2Agriculture Faculty Department of Horticulture, Van Yüzüncü Yıl University, Van 65000, Turkey; 3Lapseki Vocational School Department of Food Processing, Çanakkale Onsekiz Mart University, Çanakkale 17000, Turkey; 4Agricultural Faculty Department of Horticulture, Ege University, İzmir 35000, Turkey; 5Agricultural Faculty Department of Horticulture, Bolu Abant Izzet Baysal University, Bolu 14000, Turkey

**Keywords:** fig, storage, putrescine, bioactive compounds, organic acids

## Abstract

The storage and shelf life of the fig, which has a sensitive fruit structure, is short, and this results in excessive economic losses. In a study carried out to contribute to the solution of this problem, the effect of postharvest putrescine application at different doses (0, 0.5, 1.0, 2.0, and 4.0 mM) on fruit quality characteristics and biochemical content during cold storage in figs was determined. At the end of the cold storage, the decay rate and weight loss in the fruit were in the ranges of 1.0–1.6% and 1.0–5.0 %, respectively. The decay rate and weight loss were lower in putrescine-applied fruit during cold storage. Putrescine application had a positive effect on the changes in fruit flesh firmness values. The SSC rate of fruit varied between 14 and 20%, while significant differences in the SSC rate occurred depending on storage time and putrescine application dose. With putrescine application, the decrease in the acidity rate of the fig fruit during cold storage was smaller. At the end of the cold storage, the acidity rate was between 1.5–2.5% and 1.0–5.0. Putrescine treatments affected total antioxidant activity values and changes occurred in total antioxidant activity depending on the application dose. In the study, it was observed that the amount of phenolic acid in fig fruit decreased during storage and putrescine doses prevented this decrease. Putrescine treatment affected the changes in the quantity of organic acids during cold storage, and this effect varied depending on the type of organic acid and the length of the cold storage period. As a result, it was revealed that putrescine treatments can be used as an effective method to maintain postharvest fruit quality in figs.

## 1. Introduction

The interest in the fig, which is a symbol of a healthy and long life and is seen as being sacred, is increasing day by day because the fig is rich in fiber, amino acids, vitamins, carotenoids, minerals, antioxidants, sugars, organic acids, and sodium, because it is fat- and cholesterol-free, and due to its different evaluations [1]. Although the consumer preference for figs is fresh consumption, the fig’s postharvest life is short due to the sensitive structure of the fresh fig fruit, and therefore dry consumption of figs is greater than fresh consumption. The short storage and shelf life causes significant economic losses during marketing processes [2] and limits fig cultivation. Fig is a climacteric fruit species, and when fruit for fresh consumption are harvested at eating maturity, their ripening continues after harvest [2]. For this reason, the fig’s fruit sensory quality characteristics are quickly lost. In order to contribute to the solution of this problem, applications of Aloe vera [2], calcium chloride [3], chlorine dioxide [4], modified atmosphere packaging, a nanomist humidifier [5], and sodium hydrosulfide have been used to maintain the fruit quality of figs for a longer postharvest period. However, no study has been conducted about the application to figs of polyamines [6], which are claimed to prolong shelf life in fruit.

Polyamines such as putrescine, spermidine, and spermine, which have basic functions in living organisms, play a role in many biological processes, including cell division, cell elongation, embryogenesis, root formation, floral initiation and development, fruit development and ripening, and pollen tube growth and senescence, and in response to biotic and abiotic stress in plants [7,8,9,10,11]. Polyamines, which play a role in growth and development processes in plants, can delay ripening by slowing the respiration rate and ethylene production in fruit, reducing postharvest softening and chilling damage, and increasing resistance to diseases [12]. Previous studies have examined postharvest polyamine applications such as melatonin, putrescine, and spermidine. It has been reported that postharvest melatonin application delays senescence, increases resistance to chilling damage, and protects fruit quality by increasing resistance to diseases in fruit species such as litchi [13], mango [14], orange [15], peach [16], banana [17], sweet cherry [18], pear [19], pomegranate [20], plum [21], and kiwifruit [22] during postharvest storage. Putrescine maintains membrane integrity and delays the removal of epicuticular waxes, which play an important role in water exchange through the skin. Therefore, putrescine may be used to prolong storability and increase shelf life in fruit during cold storage.

The aim of this study, which we planned based on the potential of putrescine, was to determine the effect of putrescine applied in different doses after harvest on fruit quality characteristics and biochemical content of figs during cold storage.

## 2. Results and Discussion

### 2.1. Weight Loss, Decay Ratio, and Fruit Firmness

Weight loss, which increased proportionally with the cold storage period, was higher in the fruit of the control treatment (Figure 1). Supporting the study result, Candir et al. [23], Rasouli et al. [24], Byeon and Lee [25], and Ozturk et al. [26] reported that the weight loss resulting from the evaporation of water in the fruit by transpiration [27], and causing significant economic damage [28], increased proportionally with the storage time. Polyamines, which delay aging in fruits, decrease as maturity progresses, and this negatively affects the textural properties and storage time of fruit [29]. However, weight loss during storage can be reduced by the use of polyamines such as melatonin and putrescine, which delay cell wall degradation [30,31] and cause low respiration in fruit [32,33]. Indeed, Fawole et al. [34] suggested that putrescine treatment may reduce weight loss in fruit by consolidating the permeability of tissues and cell integrity. Considering this issue, in our study aimed at prolonging postharvest life, it was found that putrescine treatment reduced weight loss during cold storage, but the application dose was not effective (Figure 1). Similarly, Kibar et al. [35] reported that the application of putrescine to peach reduced weight loss during cold storage, due to the lower respiration rate in putrescine-applied fruit.

Significant changes in the decay rate occurred during cold storage. The decay rate increased with all applications in the first 7 days of cold storage. There were significant differences between the treatments in the measurements made during this period, where the lowest decay rate was recorded in fruit with applied doses of 1 and 4 mM of putrescine, and the highest values were recorded with the control application. The decay rate increased continuously with the control application during the cold storage period; the decay rate of the putrescine-applied fruit decreased on the 14th day of cold storage and significant differences occurred depending on the application concentration. At the end of cold storage, the highest decay rate was recorded with control and 1 mM putrescine applications, while the lowest decay rate was in 0.5 and 4 mM putrescine-applied fruit (Figure 2). Polyamines such as putrescine and spermidine, which have anti-pathogenic properties, significantly reduce rot and chilling damage in fruit during postharvest storage [36]. Previous studies have shown that polyamine applications decrease rot and chilling damage and maintain fruit quality during cold storage in peach [35,37], pomegranate [38], apricot [39], and papaya [36].

Fruit flesh firmness, which decreases as fruit maturity progresses as a result of the degradation of the cell wall components, such as pectin substances, hemicellulose, and cellulose, and the decrease in turgor pressure in the cell [40], is an important quality parameter that determines the storage potential of the fruit [41,42]. Fruit flesh firmness, which has an important effect on marketing and postharvest processes in fruit, decreases with the progress of the ripening. Previous studies have reported that the softening in the fruit flesh firmness occurred in fruit species such as jujube [43], [26], orange [24], pomegranate [23], sweet cherry [44], cherry laurel [45], and fig [46,47,48]. After harvest, the fruit flesh firmness values increased until the 7th day of cold storage, followed by a decrease, and the lowest values were obtained at the end of cold storage. The putrescine application affected the changes in the fruit flesh firmness values, with the lowest values recorded in the 2 mM putrescine-applied fruit and control applications during cold storage. At the end of cold storage, it was observed that the fruit flesh firmness was higher in the fruit treated with 0.5 mM putrescine (Figure 3). Polyamines, which delay ripening by changing the stability of the cell wall in fruit, contribute to the maintenance of fruit flesh firmness after harvest. In previous studies, it has been reported that putrescine application maintained fruit flesh firmness in fruit species such as plum [49,50], peach [35,51], and papaya [36].

### 2.2. Soluble Solids Content and Titratable Acidity

Soluble solid content and titratable acidity, which are effective in determining the quality of the fruit and its acceptability by the consumer, are important fruit characteristics that affect the postharvest storage period. As the maturity of the fruit progresses, the amount of SSC increases while the acidity rate decreases [52]. Consistent with this explanation, there was no significant change in the amount of SSC with control treatment during cold storage in the study. However, a lower SSC rate was measured in putrescine-treated fruit during cold storage compared to harvest. The significant differences in the SSC rate occurred depending on the cold storage time and putrescine application dose. For example, with applications of 0.5 and 2 mM, the SSC rate during cold storage decreased steadily. It was determined that there was a decrease in the SSC rate in the first week of storage in 1 mM putrescine-applied fruit, no change during the 2nd week, and an increase during the 3rd week. At the end of the cold storage, the lowest SSC ratio was measured in 2 mM treated fruit while the highest value was recorded in control fruit (Figure 4).

As ripening progresses, the acidity rate decreases. In the study, it was observed that there was a decrease in the titratable acidity rate in all applications during the first week of storage. At the end of cold storage (day 21), the highest titratable acidity was recorded in control fruits, and the lowest acidity was recorded in fruits treated with 1 mM putrescine. As a result, it was determined that the effect of putrescine application dose on the acidity rate of the fig fruit during storage is important (Figure 5). Putrescine, which delays ripening by reducing ethylene production in fruit, also reduces the changes in SSC and TA ratios after harvest [50]. In previous studies, it was suggested that the changes in the SSC and TA ratios were lower with putrescine application in species such as plum [50], peach [35,51,53], and papaya [36].

### 2.3. Total Phenolics and Total Antioxidant Activity

Phenolic compounds, which are the most important antioxidant components in fruit and can change depending on genetic factors, temperature, and environmental conditions during postharvest storage [54], decompose and gradually decrease with the prolongation in the storage period [55]. Putrescine application affected the total phenolic content, which generally decreased during storage, but inconsistencies occurred in this effect. The highest total phenolic value was obtained in 0.5 mM putrescine-applied fruit in the first week of cold storage, and it was determined that there was no difference between other applications. Significant differences occurred between all applications during the second week of cold storage. At the end of cold storage, the highest values were recorded with putrescine applications of 2 and 4 mM, and the lowest values were recorded in fruit treated with 1 mM of putrescine (Figure 6).

Putrescine application affected the total antioxidant activity values, which varied depending on the storage period and the applications. However, the changes in total antioxidant activity occurred depending on the putrescine application dose. At the end of cold storage, there was no difference between the total antioxidant activity values of fruit treated with 2 or 4 mM of putrescine, while the lowest values were recorded in fruit treated with 1 mM of putrescine, and the highest values were recorded in fruit of the control application (Figure 7). Indeed, Davarynejad et al. [56] reported that putrescine application was found to affect the total phenolic content and antioxidant activity of plum fruit during cold storage, with the effect varying depending on the application dose, and the most effective application dose being 4 mM. Furthermore, Kibar et al. [35] suggested that the change in total phenolic and antioxidant activity after harvest in peach was lower with putrescine application, and putrescine application of 1.6 mM was more effective.

### 2.4. Specific Phenolic Compounds

Polyphenolics are secondary metabolites that increase the quality and antioxidant properties of fruits and vegetables, such as firmness, flavor, bitterness, and color, and contribute to the defense mechanism of the plant [57]. Polyamines including melatonin, spermidine, and putrescine, which remove reactive oxygen species (ROS) and induce the gene expression of antioxidant enzymes in plants [13,58], are also used to improve fruit quality by increasing the level of some beneficial compounds, such as sucrose, natural antioxidants, phenolics, aroma components, polyphenolics, and soluble solids in fruit [19,59,60]. Polyamine application may cause an increase in the content of compounds such as amino acids, anthocyanins, phenols, and flavonoids in fruit after harvest [13,60,61]. In this study, protocatechuic acid was the individual phenolic having the highest quantity in figs, followed by chlorogenic acid, gallic acid, ferulic acid, syringic acid, and *p*-coumaric acid, in order. The quantity of these individual phenolic compounds decreased with prolonged storage time. The decrease in the quantity of phenolic acids both during and at the end of cold storage was lower in putrescine-treated fruit. It was determined that putrescine application concentration was effective in the maintenance of phenolic acids after harvest (Table 1). Rutin and catechin concentrations decreased proportionally with the storage time, while epicatechin and hydrocinnamic acid concentrations increased. It was observed that putrescine application had an effect on flavonoid compounds during cold storage, but this effect varied depending on the application concentration and flavonoid compound. Considering the measurements taken at the end of the cold storage, it was observed that putrescine application was effective in maintaining rutin and epicatechin concentrations, but had no effect on hydrocinnamic acid concentration (Table 2). Kibar et al. [35] reported that the individual phenolic contents of peach decreased during cold storage, and the putrescine application prevented the loss of these compounds, with the effect varying depending on the concentration. Putrescine application delays the biochemical changes that occur by preventing ethylene synthesis in the fruit [53].

### 2.5. Organic Acids

Organic acids are important fruit quality parameters. Their rate of change decreases with the progress of maturity in fruit and their concentration may vary depending on fruit species [62]. Fig fruit contain many organic acids, including citric acid, malic acid, fumaric acid, acetic acid, and shikimic acid, and organic acid levels vary significantly with fig cultivars, growth stages, and harvest seasons [63,64]. In this study, the highest amount of citric acid was found in figs, followed by tartaric, fumaric, and succinic acid. The changes occurred in the quantity of organic acids during cold storage (Table 3). However, Celikel and Karacali [65] reported that there was no change in the citric acid content of the fig fruit during cold storage. Putrescine application affected the changes in the quantity of organic acids during cold storage, and this effect varied depending on the type of organic acid and the length of the storage period. It can be said that putrescine application slows down the decrease in the concentration of organic acids during cold storage (Table 3). In similar studies [14,66] it was revealed that the organic acid content of the fruit was significantly maintained during storage with polyamine applications.

### 2.6. Correlation between Postharvest Putrescine Treatments and Quality Characteristics of Fruits, Organic Acids, and Phenolic Compounds by PCA

Statistical evaluation is of significant importance for the scientific evaluation of the findings obtained in research and for the correct interpretation of the results. Principal component analysis (PCA) is an important statistical method that scientifically reveals the relationship between the results of research and the effect of the applied method on the findings [18,67,68]. In this study, the effect of putrescine applications at different doses after harvest on the quality and biochemical properties of fig fruits was revealed by PCA. In the PCA performed to measure the reaction of some quality parameters of fig fruits to putrescine applications during storage, the correlation was 63.1% (PCA 1 + PCA 2) (Figure 8). In this study, it was determined that there was a positive correlation between total phenolic and total antioxidant, but there was a negative correlation between degradation rate and these biochemicals. It was observed that there was an inverse relationship between weight loss and acidity values and fruit firmness. Examination of the PCA plane shows that TA, TP, and pH were found in the first plane. On the PCA plane, acidity and weight loss were in the second region, fruit firmness was in the third region, and SSC and DR were in the fourth region. Examination of the correlation between putrescine application doses and the control group showed that the results from increasing doses of putrescine differed from those of the control group. On the PCA plane, 0.5 and 2 mM were located in the first region, 1 and 4 mM were in the third region, and the control group was grouped in the fourth region. Statistical evaluations of the storage period showed that the 7th day group formed a large intersection area with the 14th day group. It was determined that there was no clear distinction between the three storage periods (7th day, 14th day, and 21st day) and there were common intersection areas.

In the PCA analysis performed to evaluate the statistical relationship between the phenolic compound contents of fig fruits and the postharvest putrescine application doses, it was found that the correlation was 54.6% (PCA 1 + PCA 2). It was determined that there were no phenolic compounds in the first and third regions of the PCA plane. Catechin, rutin, ferulic, and protocatechuic acids were located in the second region in the PCA plane, and other phenolic compounds were located in the fourth region (Figure 9). A positive correlation between catechin, rutin, ferulic, and protocatechuic values, and a parallelism, were determined. A significantly negative correlation was observed between *o*-coumaric and catechin. Significant differences were determined between the putrescine doses and the control group. In this study, on the PCA plane, the control group was located in the second region, 0.5 and 1 mM in the first region, 2 mM in the second region, and 4 mM in the fourth region. Examination of the correlation between storage times showed that there was a large difference between the 7th day storage and the 21st day storage, and the intersection ratio was very low. Therefore, it was determined that as the storage time increased, the change in the phenolic compound content of the fig fruits was higher and there was a negative relationship between storage time and phenolic compounds.

In this study, it was observed that the organic acid content of fig fruits generally decreased during storage. According to PCA analysis, the correlation between organic acids was 88% (PCA 1 + PCA 2). This ratio largely revealed the correlation between organic acids by two main factors of principal component analysis (Figure 10). In this study, it was observed that malic acid differed from other organic acids and was located in the second region in the PCA plane. Tartaric, succinic, fumaric, and citric acid were located in the fourth region in the PCA plane. When the correlation between the control group and putrescine doses was examined, it was observed that the 0.5 mM putrescine dose was grouped in the third region and the other doses were grouped in the fourth region together with the control group. It was determined that the change in organic acids emerged clearly during the storage period and the rate of decrease in these compounds was high. It was determined that there were significant differences between the 7th day and 21st day findings in storage and that there was a low level of intersection between these two groups in the cluster analysis.

## 3. Materials and Methods

Fruit harvested in an orchard established using the “Bursa Siyahı” cultivar at a planting distance of 5 m × 5 m in Siirt in 2012 were placed in plastic boxes (Plastas, Turkey) with a capacity of 5 kg and immediately transferred to Laboratory of Horticulture Department of Siirt University (Siirt, Turkey), within 1 h using a refrigerated vehicle (12 ± 1.0 °C and 75 ± 5.0% RH). The fruits were harvested when the SSC ratio was 19%. The maturity index was added to the “Materials and Methods” section of the manuscript as shown below. The fruits were harvested when the SSC rate was 19%. Twenty fruit were used for harvest period analysis and measurements. The remaining 300 fruit were grouped into groups of 60 fruit for 5 treatments (control, 0.5 mM, 1 mM, 2 mM, 4 mM putrescine). One group was selected as the control group, and fruit of the other four groups were immersed in putrescine solution prepared at different concentrations for 15 min. According to previous research, one of the main effects of postharvest polyamines is preservation of fruit firmness. Flesh firmness augmentation and fruit softness reduction have been reported in most horticultural crops, such as strawberries [69], peach [70], and plum [50].

For each application, the fruit were put into 3 plastic containers of 1 kg to be used in analyses and measurements conducted at different periods after harvest and stored at 0 ± 0.5 °C and 90 ± 5% RH for 7, 14, and 21 days. Fruit were analyzed at the end of each storage period.

### 3.1. Weight Loss

At the beginning of cold storage, initial weights (Wi) of the fruit were determined by a digital scale with a precision of 0.01 g (Radwag, Poland). Then, on days 7, 14, and 21 of the storage, final weights (Wf) were determined. The weight loss that occurred in fruit was based on the weight at the beginning of each measurement period and determined as a percentage through the equation given below (Equation (1)):(1)$$WL=\frac{Wi-Wf}{Wi}\times 100$$

### 3.2. Decay Rate

Before the cold storage, the fruit (about 0.5 kg of fruit) were counted in each replication and the total number of fruit (TF) was determined. Then, during each measurement period, the decayed fruit (DF) in each replication were determined. If the development of mycelium on the shell occurred, the fruit were considered rotten. Finally, with the following equation (Equation (2)), the decay rate (DR, %) was detected:(2)$$DR=\frac{TF-DF}{TF}\times 100$$

### 3.3. Fruit Firmness

Five fruit from each replication were used to determine firmness. The fruit skin was cut at two different points (on the cheeks) along the equatorial part of the fruit and the firmness was determined using a penetrometer (FT–327; McCormick, WA, USA) with a 7.9 mm penetrating tip. Firmness was stated in Newtons (N).

### 3.4. Soluble Solids Content, Titratable Acidity

Soluble solids content (SSC) was determined with a digital refractometer (Atago PAL-1, Washington, USA) and recorded as a percentage (%). For titratable acidity (TA) measurement, 10 mL of distilled water was added to 10 mL of juice. Then, 0.1 N sodium hydroxide (NaOH) was added until the solution’s pH reached 8.2. Based on the amount of NaOH consumed in titration, titratable acidity was determined and stated as g malic acid/kg [34].

### 3.5. Total Phenolics and Antioxidant Capacity

During each measurement period, five fruit taken from each replication were first washed with distilled water. The fruit were homogenized by a blender (Promix HR2653 Philips, Turkey). About 30 mL of homogenate was taken and placed into a 50 mL falcon tube. The prepared tubes were kept at −20 °C until the time of analysis. Before the analyses, the frozen samples were dissolved at room temperature (21 °C). Pulp and juice were separated from each other by a centrifuge at 12,000× *g* at 4 °C for 35 min. The resultant filtrate was used to determine the content of total phenolics and antioxidant capacity. Spectrophotometric measurements for total phenolics and antioxidant capacity were performed using a UV-Vis spectrophotometer (Shimadzu, Kyoto, Japan) [71,72].

### 3.6. Specific Phenolic Compounds

Specific phenolic compounds were analyzed as follows. One gram of homogeneously selected fresh fruit samples was weighed and extracted with methyl alcohol (5 mL) in a test tube for 6 h. The extract was analyzed by high pressure liquid chromatography (HPLC) (Perkin-Elmer series 200, Norwalk, CT, USA). The HPLC system was equipped with a DAD detector (Agilent, USA) and quaternary solvent dispensing system (Series 200, analytical pump) and used at 280 nm. Analyses were separated by a chromatographic separation performed with a 250 × 4.6 mm, 4 µm ODS column (HiChrom, Bergenfield, NJ, USA). The column temperature was adjusted to 26 °C in the automatic temperature regulation system in the HPLC device. Solvent A (methanol/acetic acid/water; 10:2:28) and solvent B (methanol/acetic acid/water; 90:2:8) were used as the mobile phase (Table 4). The mobile phase flow rate was maintained at 1 mL per minute and 20 μL of the sample was injected, and in the light of the results of the obtained peak areas, the contents of phenolic compounds were expressed as mg/100 g FW [73].

### 3.7. Organic Acids

Extraction of organic acids in fresh and dried samples was carried out with a modification of the method reported by Bevilacqua and Califano [74]. A quantity of 10 g of sample was placed in centrifuge tubes and then 10 mL of 0.009 N sulfuric acid was added to the samples and homogenized. The samples were mixed for 1 h and centrifuged at 14,000 rpm for 15 min. The liquid (supernatant) remaining at the top of the centrifuge tube was filtered through filter paper, then passed through a 0.45 μm membrane filter (Millipore Millex-HV Hydrophilic PVDF, Millipore, USA) and finally through a SEP-PAK C18 cartridge. It was injected into the HPLC (Agilent HPLC 1100 series G 1322 A, Germany) device and the separations were performed on the appropriate column (Aminex HPX—87 H, 300 mm × 7.8 mm). Organic acids were determined at wavelengths of 214 and 280 nm. A quantity of 0.009 N H_2_SO_4_ solution was used as mobile phase [74].

### 3.8. Statistical Analysis

One-way ANOVA was used to analyze the effect of postharvest putrescine applications on the investigated properties of fig fruits. Differences among means were evaluated by the Tukey HSD test and the significance was accepted at the *p* < 0.05 level. The ANOVA analyses were performed using JMP 16 (SAS Institute Inc., Cary, NC, USA). Principal component analysis (PCA) was performed using the JMP 16 (SAS Institute Inc., Cary, NC, USA) program to determine the relationship between postharvest applications of fig fruits, and their physicochemical properties and storage times.

## 4. Conclusions

Putrescine application reduced the weight loss and decay rate in storage while it delayed the softening of the fruit. Significant differences in SSC rate occurred depending on storage time and putrescine application dose. At the end of the cold storage, the lowest SSC ratio was measured in 2 mM treated fruit, while the highest value was recorded in control fruit. With putrescine application, the decrease in the acidity rate of the fig fruit during storage was smaller. Putrescine application affected the total phenolic content, which generally decreased during storage, but there were inconsistencies in this effect. Putrescine application affected the total antioxidant activity values and changes occurred in total antioxidant activity depending on the application dose. The decrease in the quantity of phenolic acids both during and at the end of the cold storage was smaller in putrescine-treated fruit. It was observed that putrescine application had an effect on flavonoid compounds during cold storage, but this effect varied depending on the application concentration and flavonoid compound. Putrescine application affected the changes in the quantity of organic acids during cold storage, and this effect varied depending on the type of organic acid and storage time. As a result, it was revealed that putrescine application can be used as an effective method to preserve fruit quality after harvest in figs.

## Figures and Tables

**Figure 1 plants-12-01291-f001:**
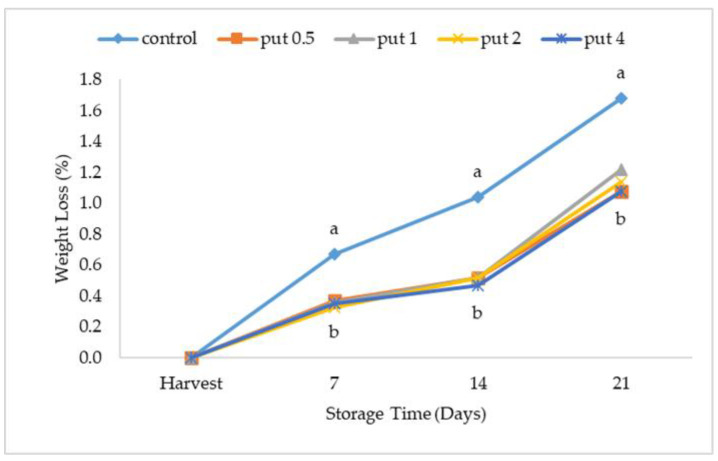
Effect of putrescine treatment on weight loss of fig fruit during cold storage. Put 0.5, 1, 2 and 4 indicates putrescine doses of 0.5 mM, 1 mM, 2 mM, and 4 mM, respectively. According to the Tukey test, *p* < 0.05 does not differ between the same letters in the same harvest period.

**Figure 2 plants-12-01291-f002:**
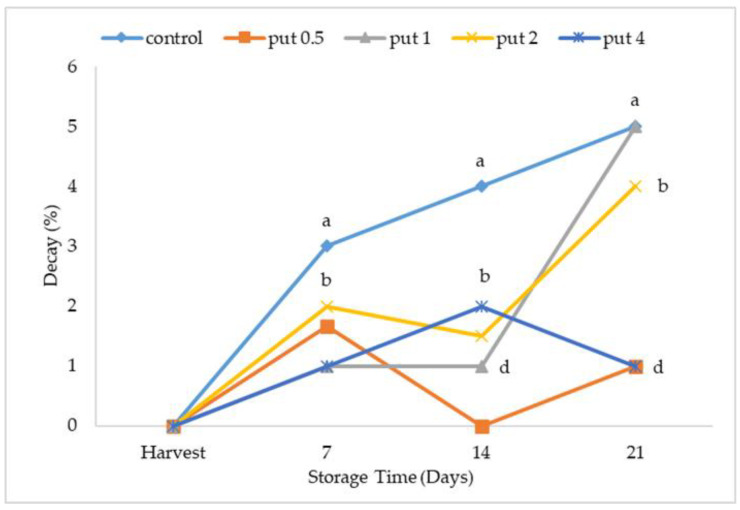
Effect of putrescine treatment on decay ratio of fig fruit during cold storage. Put 0.5, 1, 2 and 4 indicates putrescine doses of 0.5 mM, 1 mM, 2 mM, and 4 mM, respectively. According to the Tukey test, *p* < 0.05 does not differ between the same letters in the same harvest period.

**Figure 3 plants-12-01291-f003:**
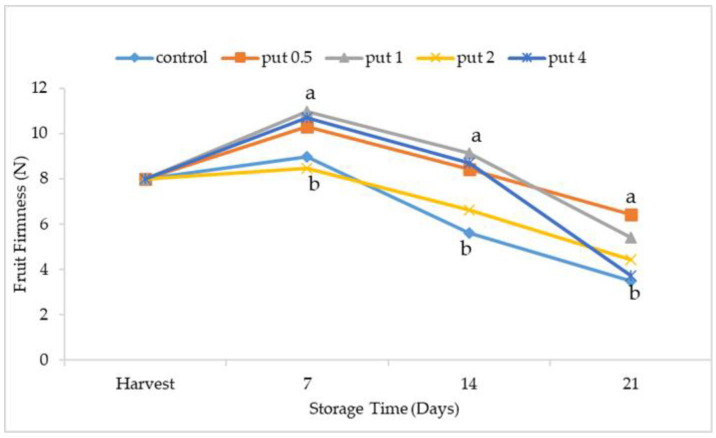
Effect of putrescine treatment on fruit firmness of fig fruit during cold storage. Put 0.5, 1, 2 and 4 indicates putrescine doses of 0.5 mM, 1 mM, 2 mM, and 4 mM, respectively. N: Newton. According to the Tukey test, *p* < 0.05 does not differ between the same letters in the same harvest period.

**Figure 4 plants-12-01291-f004:**
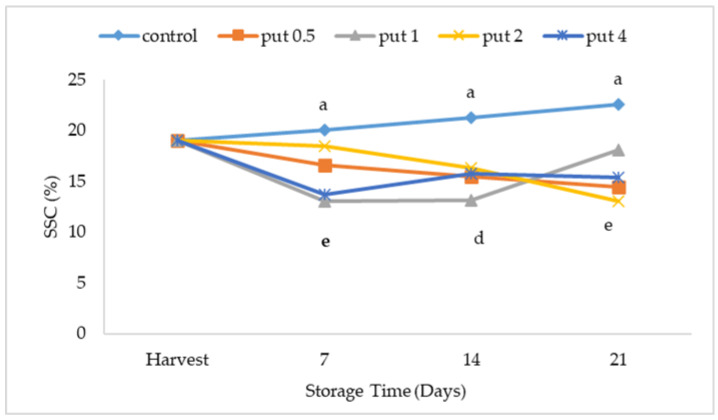
Effect of putrescine treatment on soluble solids content (SSC) of fig fruit during cold storage. Put 0.5, 1, 2 and 4 indicates putrescine doses of 0.5 mM, 1 mM, 2 mM, and 4 mM, respectively. According to the Tukey test, *p* < 0.05 does not differ between the same letters in the same harvest period.

**Figure 5 plants-12-01291-f005:**
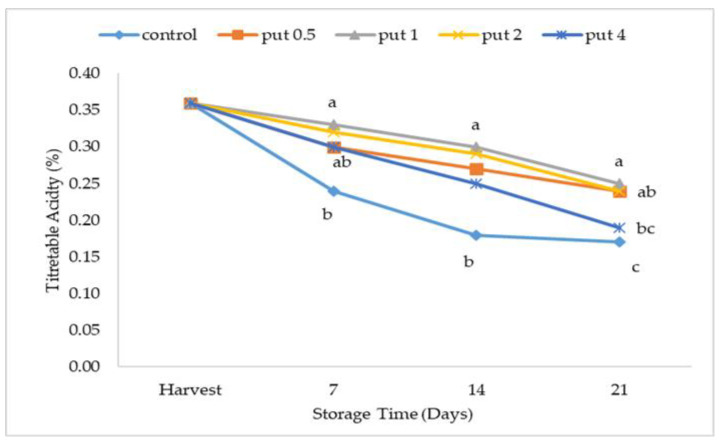
Effect of putrescine treatment on titratable acidity of fig fruit during cold storage. Put 0.5, 1, 2 and 4 indicates putrescine doses of 0.5 mM, 1 mM, 2 mM, and 4 mM, respectively. According to the Tukey test, *p* < 0.05 does not differ between the same letters in the same harvest period.

**Figure 6 plants-12-01291-f006:**
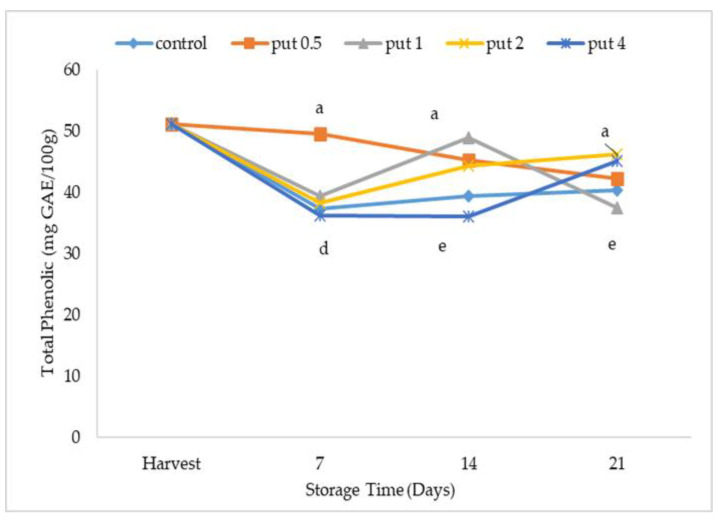
Effect of putrescine treatment on total phenolics of fig fruit during cold storage. Put 0.5, 1, 2 and 4 indicates putrescine doses of 0.5 mM, 1 mM, 2 mM, and 4 mM, respectively. According to the Tukey test, *p* < 0.05 does not differ between the same letters in the same harvest period.

**Figure 7 plants-12-01291-f007:**
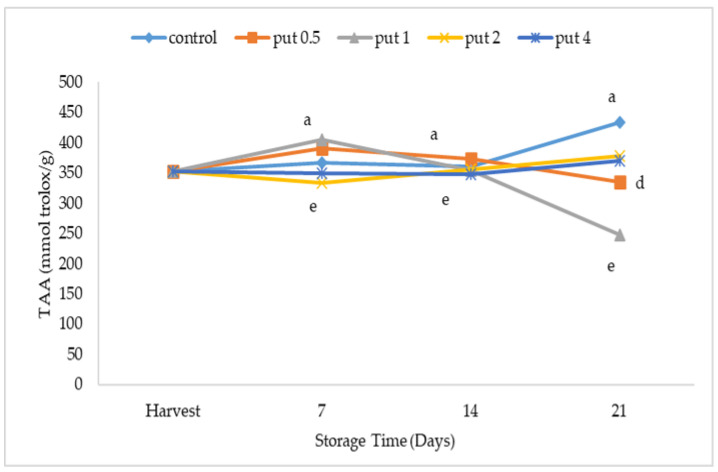
Effect of putrescine treatment on total antioxidant activity (TAA) of fig fruit during cold storage. put 0.5, 1, 2 and 4 indicates putrescine doses of 0.5 mM, 1 mM, 2 mM, and 4 mM, respectively. According to the Tukey test, *p* < 0.05 does not differ between the same letters in the same harvest period.

**Figure 8 plants-12-01291-f008:**
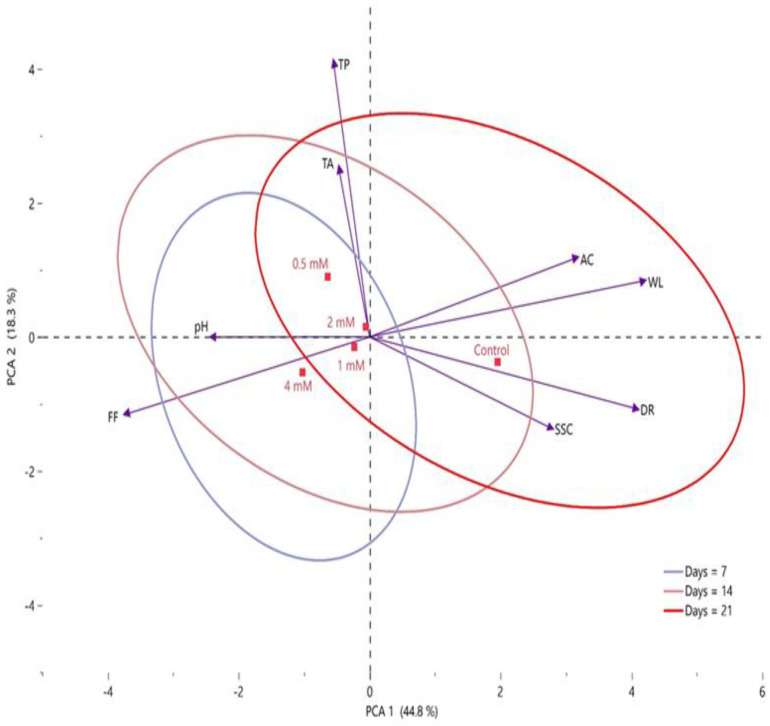
Determination of the effects of putrescine doses (control, 0.5 mM, 1 mM, 2 mM, and 4 mM) on agromorphological properties during storage by PCA. WL: Weight loss, SSC: Soluble solid contents, AC: Acidity, TP: Total phenolic, TA: Total antioxidant, DR: Decay Rate, FF: Fruit Firmness.

**Figure 9 plants-12-01291-f009:**
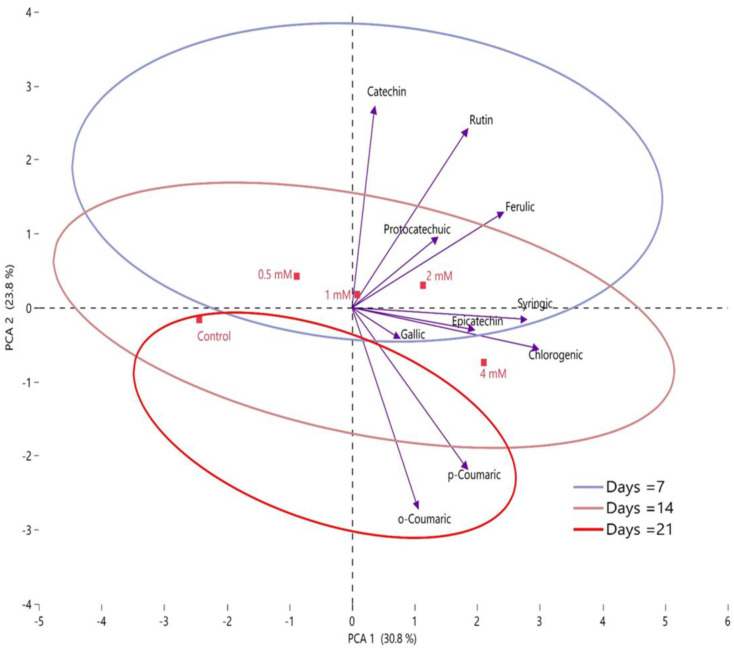
Determination of the effects of putrescine doses (control, 0.5 mM, 1 mM, 2 mM, and 4 mM) on phenolic compounds contents during storage by PCA.

**Figure 10 plants-12-01291-f010:**
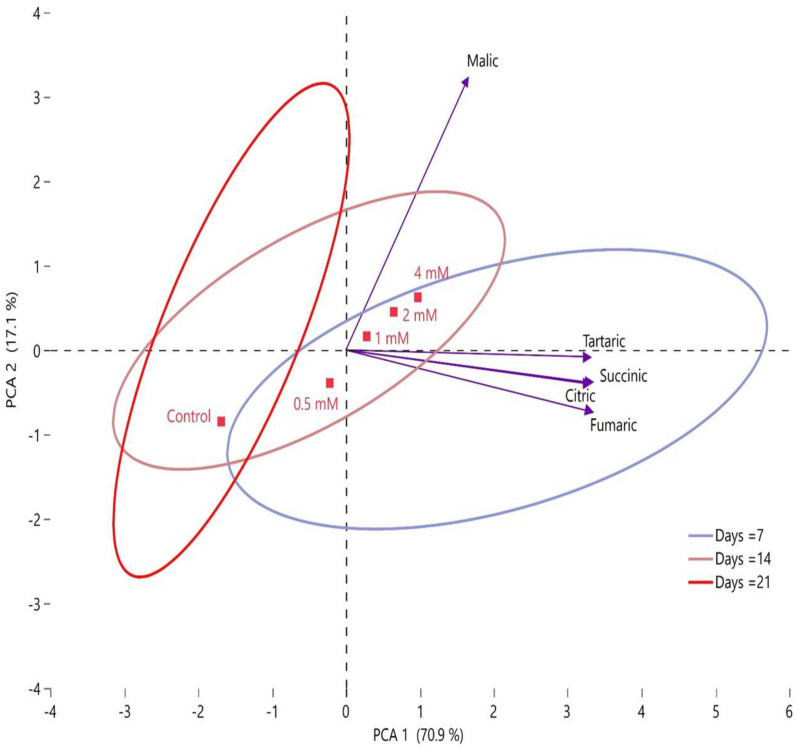
Determination of the effects of putrescine doses (control, 0.5 mM, 1 mM, 2 mM, and 4 mM) on organic acid contents during storage by PCA.

**Table 1 plants-12-01291-t001:** Effect of putrescine treatment on phenolic acids of fig fruit during cold storage.

Phenolic Compounds (mg/100 g FW)	Putrescine Treatment (mM)	Storage Time (Days)			
		Harvest	7	14	21
Gallic acid	Control	1.52	1.50 bc	1.31 b	1.40 c
	0.5		1.37 c	1.36 b	1.70 bc
	1		2.25 a	1.93 a	1.77 ba
	2		1.70 ac	0.64 c	2.06 a
	4		1.03 ba	1.57 ba	1.75 ba
*p*-Coumaric acid	Control	0.18	0.10 a	0.13 c	0.20 b
	0.5		0.19 a	0.18 c	0.40 ba
	1		0.23 a	0.31 c	0.71 a
	2		0.25 a	0.65 b	0.74 a
	4		0.10 a	0.89 a	0.75 a
Chlorogenic acid	Control	2.08	1.25 d	1.51 d	2.15 a
	0.5		1.31 d	1.72 dc	2.72 a
	1		2.74 c	1.97 c	2.32 a
	2		3.22 b	2.63 b	2.80 a
	4		3.98 a	4.04 a	2.82 a
Syringic acid	Control	0.28	0.17 b	0.27 c	0.09 b
	0.5		0.23 b	0.32 c	0.18 b
	1		0.31 b	0.82 b	0.56 a
	2		0.33 b	0.85 b	0.66 a
	4		1.88 a	1.66 a	0.15 b
Ferulic acid	Control	0.41	1.16 c	1.34 c	0.37 d
	0.5		1.82 b	1.94 b	0.77 c
	1		1.91 ba	1.90 b	0.88 cb
	2		1.93 ba	2.05 b	1.62 a
	4		2.17 a	2.85 a	1.22 b
Protocatechuic acid	Control	3.02	2.16 b	1.59 b	2.73 cb
	0.5		4.69 a	2.86 ba	3.18 b
	1		3.15 b	2.49 ba	5.07 a
	2		5.50 a	3.48 a	2.29 c
	4		4.49 a	2.62 ba	4.62 a
*o*-Coumaric acid	Control	0.35	0.04 c	0.08 a	0.33 a
	0.5		0.07 cb	0.14 a	0.36 a
	1		0.08 cb	0.16 a	0.38 a
	2		0.12 c	0.20 a	0.52 a
	4		0.34 a	0.26 a	0.53 a

Means in columns with the same letter do not differ according to Tukey’s test at *p* < 0.05.

**Table 2 plants-12-01291-t002:** Effect of putrescine treatment on flavonoid and contents of fig fruits during cold storage.

Flavonoid Acid Contents (mg/100 g FW)	Putrescine Treatment (mM)	Storage Time (Days)			
		Harvest	7	14	21
Rutin	Control	5.88	5.79 d	2.20 d	1.36 c
	0.5		7.64 c	3.71 c	2.70 b
	1		8.57 cb	3.72 c	3.50 b
	2		9.01 b	4.62 b	3.11 b
	4		10.56 a	5.38 a	4.88 a
Catechin	Control	2.31	0.73 cb	2.25 c	0.14 b
	0.5		076 cb	0.49 bac	0.49 a
	1		0.85 b	0.56 ba	0.46 a
	2		1.47 a	0.70 a	0.47 a
	4		0.34 c	0.26 bc	0.09 b
Epicatechin	Control	0.71	0.17 d	0.33 c	0.67 c
	0.5		0.66 c	2.34 b	1.62 b
	1		1.36 b	1.78 b	1.70 b
	2		1.81 a	4.03 a	2.26 a
	4		1.29 b	2.22 b	0.72 c

Means in columns with the same letter do not differ according to Tukey’s test at *p* < 0.05.

**Table 3 plants-12-01291-t003:** Effect of putrescine treatment on organic acids (mg/100 g FW) contents of fig fruits during cold storage.

Organic Acids	Putrescine Treatment (mM)	Storage Time (Days)			
		Harvest	7	14	21
Citric acid	Control	14.21	12.20 b	7.19 b	8.02 b
	0.5		19.00 a	8.25 b	11.13 a
	1		19.05 a	12.14 a	12.22 a
	2		20.02 a	12.67 a	12.36 a
	4		19.96 a	13.50 a	8.56 b
Succinic acid	Control	0.61	0.55 b	0.12 c	0.05 c
	0.5		0.74 b	0.18 c	0.06 c
	1		1.56 a	0.25 c	0.25 a
	2		1.62 a	0.44 b	0.28 a
	4		1.30 a	0.69 a	0.17 b
Fumaric acid	Control	1.03	0.92 b	0.70 b	0.30 c
	0.5		1.86 a	1.19 ba	0.67 a
	1		1.87 a	0.86 ba	0.69 a
	2		1.92	0.93 ba	0.43 c
	4		1.99	1.28 a	0.33 c
Tartaric acid	Control	12.21	11.29 c	9.14 b	7.66 c
	0.5		13.35 b	11.28 b	9.22 ba
	1		13.39 b	10.74 b	8.36 bc
	2		14.89 b	11.25 b	9.40 ba
	4		17.57 a	14.07 a	10.14 a

Means in columns with the same letter do not differ according to Tukey’s test at *p* < 0.05.

**Table 4 plants-12-01291-t004:** Gradient elution and time program for HPLC.

Time (min)	Dissolvent A (%)	Dissolvent B (%)
0	100	0
15	85	15
25	50	50
35	15	85
45	0	100

## Data Availability

All-new research data were presented in this contribution.

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
