# Peer review of "Impact of Postharvest Putrescine Treatments on Phenolic Compounds, Antioxidant Capacity, Organic Acid Contents and Some Quality Characteristics of Fresh Fig Fruits during Cold Storage"

_plants, 2023, doi:10.3390/plants12061291_

Round 1

Reviewer 1 Report

This manuscript advances the knowledge of the effectiveness of putrescine treatment to decrease the decay and weight loss of figs during cold storage. Also, it describes the content of the main polyphenolic compounds and organic acids in these fruits and their behavior during storage. The main weakness would be that the analyses do not show in a clear way how putrescine treatment modulates the contents of these bioactive compounds and the other response variables. I believe that a factorial analysis would be a good strategy to clearly identify if the array of response variables evaluated is significantly modulated by the concentration of putrescine (factor A) time of storage (factor B) or its combination (AxB). Also, I noticed some inconsistencies that require attention before further consideration. Please see specific comments below.

The abstract lack an introductory sentence, for instance, it could be indicated that figs are highly perishable fruits with a short shelf life, and since polyamines have been demonstrated to improve fruit’s postharvest life this study was conducted to….  

The introduction should be improved and enriched to provide a broader perspective of the fig perishability and the potential of polyamines to improve its quality during postharvest storage.

The days of storage in line 337 do not correspond to the rest of the manuscript.  

Line 72. It is not clear why the authors indicate that the application dose was not effective.

I think the authors are using the terms “rate” and “ratio” in the wrong way when referring to the levels of SSC or other parameters. For instance, SSC/Acidity ratio is useful for assessing the maturity index of fruits. What was the maturity index at the beginning of the experiment? was it calculated?

Indicate the separation methods for HPLC analyses, including time and gradient of mobile phases.

Line 66. Include Patel et al. reference in an adequate format

Line 88. Typo “yhe”

Line 382. Typo “clone”

Line 402. Replace typical with normal. The Kolmogorov-Smirnov test is used when n>50. In case, use an appropriate normality test for a small sample size to corroborate results.

Figures. Replace kontrol with control. In some cases, the literals are confusing since it is not clear which treatments they refer to.

Figures 2 and 3. Delete the number above points in the harvest column

Figure 5. The most common behavior in stored climacteric fruit is to decrease the acidity level. Are there studies to compare or support the findings in this study regarding the increase of titratable acidity during cold storage?  In fact, when looking at Table 3 information, it is observed that all organic acids decreased during storage in control fruit, and values at day 21 of storage are lower than those at harvest, which altogether does not agree with this figure. Please revise.

Fig. 7. Replace Troloks with Trolox. Since the writing in the y-axis is uppercased is not clear if MMOL is micro or milli mol.

Table 2. Typo “hidrocinomic”

Line 401. The statistical section does not indicate if and how correlations and PCA analyses were performed.

In the results and discussion section, I would suggest first describing the findings of the study and then addressing the discussion and comparison with other research studies.

Author Response

Manuscript ID: plants-2218928

Title:
Impact of postharvest putrescine treatments on phenolic compounds, antioxidant capacity, organic acid contents and some quality characteristics of fresh fig fruits during cold storage

Authors: Emine Kucuker, Erdal Aglar, Mustafa Sakaldas, Fatih Sen, Muttalip Gundogdu

Dear Editor,

We have received the results of the evaluation process for our article, " Impact of postharvest putrescine treatments on phenolic compounds, antioxidant capacity, organic acid contents and some quality characteristics of fresh fig fruits during cold storage" regarding publication in your Plant Journal.

We are happy that our manuscript will be reconsider for publication in your Journal.

We are thanks to Reviewer #1 for valuable comments and corrections.

We have carefully reviewed each of the points raised in the evaluation process, and Our answers to Editor and Reviewer #1 were indicated red  in the manuscript, respectively. However, the language was checked, and corrections were made. The corrections were indicated as green in the text.

Further details are given below. If you have any question, please contact me by email.

With warmest personal regards,

Changes Made:

Reviewer#1

Q1) The abstract lack an introductory sentence, for instance, it could be indicated that figs are highly perishable fruits with a short shelf life, and since polyamines have been demonstrated to improve fruit’s postharvest life this study was conducted to

A1) The following paragraph was added to the abstract.

The storage and shelf life of the fig, which has a sensitive fruit structure, is short, and this results in too much economic losses. In the study carried out to contribute to the solution of this problem, the effect of postharvest putrescine application at different doses (0, 0.5, 1.0, 2.0 and 4.0 mM) on fruit quality characteristics and biochemical content during cold storage in figs was determined.

Q2) The introduction should be improved and enriched to provide a broader perspective of the fig perishability and the potential of polyamines to improve its quality during postharvest storage.

A2) The following paragraph was added to the introduction. The references were reedited both in the text and in the references part.

“Polyamines, which play a role in growth and development processes in plants, can delay ripening by slowing respiration rate and ethylene production in fruit, reduce postharvest softening and chilling damage, and increase resistance to diseases [12]. In previous studies, it has been reported that the postharvest polyamine applications such as melatonin, putrescine and spermidine postharvest melatonin application delays senescence and increases resistance to chilling damage and protects fruit quality by increasing resistance to diseases in fruit species such as litchis [13], mango [14], orange [15], peach [16], banana [17], sweet cherry [18], pear [19], pomegranate [20], plum [21]and kiwifruit [22] in postharvest storage.”

Q3) The days of storage in line 337 do not correspond to the rest of the manuscript

A3) Line 337 was revised.

Q4) Line 72. It is not clear why the authors indicate that the application dose was not effective.

A4) Manuscript was revised. “The effect of application dose are significant for their using in the practice. “

Q5) I think the authors are using the terms “rate” and “ratio” in the wrong way when referring to the levels of SSC or other parameters. For instance, SSC/Acidity ratio is useful for assessing the maturity index of fruits. What was the maturity index at the beginning of the experiment? was it calculated?

A5) The fruits were harvested when the SSC ratio was 19%. The maturity index was added to Materials and Methods" section of the Manuscript as shown below.

“The fruits were harvested when the SSC rate was 19%.”

Q6) Indicate the separation methods for HPLC analyses, including time and gradient of mobile phases

A6) Separation methods for HPLC analysis, including time and gradient of mobile phases, were added to the "Materials and Methods" section of the Manuscript as shown below.

“Solvent A (methanol/acetic acid/water; 10:2:28) and solvent B (methanol/acetic acid/water; 90:2:8) were used as the mobile phase (Table 4). The mobile phase flow rate was maintained at 1 mL per minute and 20 μL of the sample was injected and expressed in g 100g-1 in light of the results of the peak areas obtained.”

Table 4. Gradient elution and time program for HPLC.

Time (min)

Dissolvent A (%)

Dissolvent B (%)

0

100

0

15

85

15

25

50

50

35

15

85

45

0

100

Q7) Line 66. Include Patel et al. reference in an adequate format

A7) The mistake was corrected by deleting Patel et al.

Q8) Line 88. Typo “yhe”

A8) “yhe” was corrected as “the”

Q9) Line 382. Typo “clone”

A9) “clone”was corrected as “column”

Q10) Line 402. Replace typical with normal. The Kolmogorov-Smirnov test is used when n>50. In case, use an appropriate normality test for a small sample size to corroborate results.

A10) The "Statistical analysis" part of the article was revised, and the revisions were added to the article as follows.

“One-way ANOVA was used to analyze the effect of postharvest putrescine applica-tions on the investigated properties of fig fruits. Differences among means were evaluated by the Tukey HSD test and the significance was accepted at P < 0.05 level. The ANOVA analysis were performed using JMP 16 (SAS Institute Inc., Cary, NC). Principal component analysis (PCA) was performed using the JMP 16 (SAS Institute Inc., Cary, NC) program to determine the relationship between postharvest applications of fig fruits, and their physi-cochemical properties and storage times.”

Q11) Figures. Replace kontrol with control. In some cases, the literals are confusing since it is not clear which treatments they refer to.

A11) Kontrol in the figures was corrected control (0.0)

Q12) Figures 2 and 3. Delete the number above points in the harvest column

A12) The number above points in the harvest column in Figures 2 and 3.  were deleted.

Q13) Figure 5. The most common behavior in stored climacteric fruit is to decrease the acidity level. Are there studies to compare or support the findings in this study regarding the increase of titratable acidity during cold storage?  In fact, when looking at Table 3 information, it is observed that all organic acids decreased during storage in control fruit, and values at day 21 of storage are lower than those at harvest, which altogether does not agree with this figure. Please revise.

A13). Figure 5 was corrected as follows and revisions was made on the Manuscript.

As ripening progresses, the acidity rate decreases. In the study, it was observed that there was a decrease in the titratable acidity rate in all applications in the first week of storage. At the end of cold storage (day 21), the highest titratable acidity was recorded in control fruits, and the lowest acidity was recorded in fruits treated with 1 mM putrescine. As a result, it was determined that the effect of putrescine application dose on the acidity rate of the fig fruit during storage is important (Figure 5).”

Q14) Fig. 7. Replace Troloks with Trolox. Since the writing in the y-axis is uppercased is not clear if MMOL is micro or milli mol.

A14) “Troloks”was corrected as “Trolox”. MMOL was changed as MICROMOL

Q15) Table 2. Typo “hidrocinomic”

A15) “hidrocinomic” in table 2 and in the text was corrected as “hydrocinnamic acid”

Q16) Line 401. The statistical section does not indicate if and how correlations and PCA analyses were performed.

A16) The "Statistical analysis" part of the article was revised, and the revisions were added to the article as follows.

“Principal component analysis (PCA) was performed using the JMP 16 (SAS Institute Inc., Cary, NC) program to determine the relationship between postharvest applications of fig fruits, and their physi-cochemical properties and storage times.”

Q17) In the results and discussion section, I would suggest first describing the findings of the study and then addressing the discussion and comparison with other research studies.

A17) Manuscript was revised. The results and discussion section was planned as the importance of the fruit characteristics and the problem, the results obtained, the related studies and the explanation of the reason for the result.

Reviewer 2 Report

Paper:

Impact of postharvest putrescine treatments on phenolic compounds, antioxidant capacity, organic acid contents and some quality characteristics of fresh fig fruits during cold storage

by Kucuker et al., Plants

General comments:

the present work dealt with the evaluation of the effect of postharvest putrescine treatment on fig fruits quality characteristics and biochemical content during refrigerated storage. The authors focused on the biochemical aspects such as phenolic compounds, antioxidant capacity and organic acid contents. Although the topic falls within the topics of Plants journal this is a typical post-harvest study like numerous similar studies already published and the paper showed several weaknesses.

Major comment:

·      In a study like this that were evaluated the quality characteristics is mandatory to investigate the evolution of microbiological parameters in particular regarding the main spoilage and pathogenic microorganisms associated to fruits and vegetables.

·      A sensory evaluation of the fruits during the refrigerated storage of fig fruits is mandatory to evaluate the differences among control and treated fruits in term of organoleptic characteristics.

Specific comment:

  • In general, the abstract should be more descriptive, reporting the main numerical data (percentages, levels, concentrations etc.). As is, the abstract does not stand alone.
  • The introduction section is realy poor. In literature there are some published paper regarding the application of putrescine to improve the quality of fruits and vegetables.
  • Line 41: please add an international references.
  • Line 328-330: Could you explain why were made the immersion of the fruit in a solution of putrescine; in my opinion would be better applied this solutions as edible coating. However an international refereces for this metodology applied is mandatory.

There are some other minor points, but I think that there is no point in mentioning them at this stage. In my opinion, the paper is not ready to be published in Plants several data should be provided. My response is rejection.

Author Response

Manuscript ID: plants-2218928

Title: Impact of postharvest putrescine treatments on phenolic compounds, antioxidant capacity, organic acid contents and some quality characteristics of fresh fig fruits during cold storage

Authors: Emine Kucuker, Erdal Aglar, Mustafa Sakaldas, Fatih Sen, Muttalip Gundogdu

Dear Editor,

We have received the results of the evaluation process for our article, " Impact of postharvest putrescine treatments on phenolic compounds, antioxidant capacity, organic acid contents and some quality characteristics of fresh fig fruits during cold storage" regarding publication in your Plant Journal.

We are happy that our manuscript will be reconsider for publication in your Journal.

We are thanks to Reviewer #2 for valuable comments and corrections.

We have carefully reviewed each of the points raised in the evaluation process, and Our answers to Editor and Reviewer #2 were indicated blue in the manuscript, respectively. However, the language was checked, and some corrections were made. The corrections were indicated as green in the text.

Further details are given below. If you have any question, please contact me by email.

With warmest personal regards,

Changes Made:

Reviewer#2

Q1) In general, the abstract should be more descriptive, reporting the main numerical data (percentages, levels, concentrations etc.). As is, the abstract does not stand alone.

A1) The storage and shelf life of the fig, which has a sensitive fruit structure, is short, and this results in too much economic losses. In the study carried out to contribute to the solution of this problem, the effect of postharvest putrecine application at different doses (0, 0.5, 1.0, 2.0 and 4.0 mM) on fruit quality characteristics and biochemical content during cold storage in figs was determined. At the end of the cold storage, the decay rate and weight loss in the fruit were between 1.0-1.6% and 1.0-5.0 respectively. The decay rate and weight loss were lower in putrescine-applied fruit during cold storage. Putrescine application had a positive effect on the changes in fruit flesh firmness values. The SSC rate of fruit varied on between 14-20 % while the significant differences in SSC rate occurred to depending on storage time and putrescine application dose. With putrescine application, the decreasing in the acidity rate of the fig fruit during cold storage was lower. At the end of the cold storage, the acidity rate was between 1.5-2.5 % and 1.0-5.0.

Q2) The introduction section is realy poor. In literature there are some published paper regarding the application of putrescine to improve the quality of fruits and vegetables.

A2) The following paragraph was added to the introduction. The references were reedited both in the text and in the references part.

Polyamines, which play a role in growth and development processes in plants, can delay ripening by slowing respiration rate and ethylene production in fruit, reduce postharvest softening and chilling damage, and increase resistance to diseases [12]. In previous studies, it has been reported that the postharvest polyamine applications such as melatonin, putrecine and spermidine postharvest melatonin application delays senensence and increases resistance to chilling damage and protects fruit quality by increasing resistance to diseases in fruit species such as litchis [13], mango [14], orange [15], peach [16], banana [17], sweet cherry [18], pear [19], pomegranate [20], plum [21]and kiwifruit [22] in postharvest storage.

Q3) Line 41: please add an international references

A3) References were added.

Q4) Line 328-330: Could you explain why were made the immersion of the fruit in a solution of putrescine; in my opinion would be better applied this solutions as edible coating. However an international refereces for this metodology applied is mandatory.

A4) The reason why the fruit is immersed in putrescine solution is explained with the literature as follows and added to the Materials and Methods.

“In researches, one of the main effects of postharvest polyamines is to preserve fruit firmness. Flesh firmness augment and fruit softness reduction have been reported in most horticultural crops, such as strawberries (Khosroshahi et al., 2007), peach (Bregoli et al., 2002) and plum (Serrano et al., 2003). “

Round 2

Reviewer 1 Report

Good job improving the paper. 

Author Response

Manuscript ID: plants-2218928

Title:
Impact of postharvest putrescine treatments on phenolic compounds, antioxidant capacity, organic acid contents and some quality characteristics of fresh fig fruits during cold storage

Authors: Emine Kucuker, Erdal Aglar, Mustafa Sakaldas, Fatih Sen, Muttalip Gundogdu

Dear Editor,

We have received the results of the evaluation process for our article, " Impact of postharvest putrescine treatments on phenolic compounds, antioxidant capacity, organic acid contents and some quality characteristics of fresh fig fruits during cold storage" regarding publication in your Plant Journal.

We are happy that our manuscript will be reconsider for publication in your Journal.

We are thanks to Reviewer #1 for their valuable comments and corrections.

We have carefully reviewed each of the points raised in the evaluation process, and Our answers to Editor and Reviewer #1 were indicated red in the manuscript, respectively.

Further details are given below. If you have any question, please contact me by email.

With warmest personal regards,

Changes Made:

Reviewer#1

Q1) Good job improving the paper.

A1) Many thanks for Reviewer#1's valuable comments.

Reviewer 2 Report

Although the general quality of the manuscript has been improved in my opinion the article showed some limitation. The authors conclude the manuscript with the affermation that on the base of the results the application of of putrescine can be used as an effective method to preserve fruit quality after harvest in figs. In my opininion this affermation is very risky.  In a study like this that were evaluated the quality characteristics is mandatory to investigate the evolution of microbiological parameters in particular regarding the main spoilage and pathogenic microorganisms associated to fruits and vegetables. Furthermore, a sensory evaluation of the fruits during the refrigerated storage of fig fruits is mandatory to evaluate the differences among control and treated fruits in term of organoleptic characteristics.

I had already highlighted these critical issues but the authors did not provide any explanation.

Author Response

Manuscript ID: plants-2218928

Title: Impact of postharvest putrescine treatments on phenolic compounds, antioxidant capacity, organic acid contents and some quality characteristics of fresh fig fruits during cold storage

Authors: Emine Kucuker, Erdal Aglar, Mustafa Sakaldas, Fatih Sen, Muttalip Gundogdu

Dear Editor,

We have received the results of the evaluation process for our article, " Impact of postharvest putrescine treatments on phenolic compounds, antioxidant capacity, organic acid contents and some quality characteristics of fresh fig fruits during cold storage" regarding publication in your Plant Journal.

We are happy that our manuscript will be reconsider for publication in your Journal.

We are thanks to Reviewer #2 for valuable comments and corrections.

We have carefully reviewed each of the points raised in the evaluation process, and Our answers to Editor and Reviewer #2 were indicated blue in the manuscript.

Further details are given below. If you have any question, please contact me by email.

Best Regards,

Changes Made:

Reviewer#2

Q1) Although the general quality of the manuscript has been improved in my opinion the article showed some limitation. The authors conclude the manuscript with the affermation that on the base of the results the application of of putrescine can be used as an effective method to preserve fruit quality after harvest in figs. In my opininion this affermation is very risky.  In a study like this that were evaluated the quality characteristics is mandatory to investigate the evolution of microbiological parameters in particular regarding the main spoilage and pathogenic microorganisms associated to fruits and vegetables. Furthermore, a sensory evaluation of the fruits during the refrigerated storage of fig fruits is mandatory to evaluate the differences among control and treated fruits in term of organoleptic characteristics.

A1) The storage and shelf life of the fig, which has a sensitive fruit structure, is short, and this results in too much economic losses. In the study carried out to contribute to the solution of this problem, the effect of postharvest putrescine application at different doses on fruit quality characteristics and biochemical content during cold storage in figs was determined. During post-harvest storage, decay can occurs in fruits as a result of microbiological activities. In this context, the decay rate was calculated in the study and the evaluations were stated in the Manuscript as follows.

“The significant changes in the decay rate occurred during cold storage. The decay rate increased with all applications in the first 7 days of the cold storage. There were sig-nificant differences between the treatments in the measurements made during this period whereas the lowest decay rate was recorded in fruit with 1 and 4 mM putrescine applied, and the highest values were recorded with the control application. The decay rate in-creased continuously with the control application during the cold storage period, the de-cay rate of the putrescine-applied fruit decreased on 14th day of the cold storage and the significant differences occurred to depending on the application concentration. At the end of the cold storage, the highest decay rate was recorded with control and 1 mM putrescine applications while the lowest decay rate was in 0.5 and 4 mM putrescine- applied fruit (Figure 2). Polyamines such as putrescine and spermidine, which have anti-pathogenic properties, significantly reduce rot and chilling damage in fruit in postharvest storage [36]. Previous studies have shown that polyamine applications decrease the rot and chilling damage and maintain fruit quality during cold storage in peach [35, 37], pomegranate [38], apricot [39] and papaya [36].”

Additionally, during post-harvest storage, decay can occurs in fruits as a result of microbiological activities. In this context, the decay rate was calculated in the study and the evaluations were stated in the Manuscript. Also, microbiological analysis and sensory analyzes were not performed in this study. In many studies, it has been reported that putrescine applications in fruits preserve fruit quality after harvest. Accordingly, it has been suggested to use polyamines in post-harvest applications (Khosroshahi et al., 2007), peach (Bregoli et al., 2002) and plum (Serrano et al., 2003). Therefore, findings in this direction were also obtained in our study.

“69. Khosroshahi, M.R.Z., Esna-Ashari, M., Ershadi, A. Effect of exogenous putrescine on post-harvest life of strawberry (Fragaria ananassa Duch.) fruit, cultivar Selva. Sci. Hortic. 2007, 114, 27-32.

  1. Bregoli, A.M., Scaramagli, S., Costa, G., Sabatini, E., Ziosi, V., Biondi, S., Torrigiani, P. 2002. Peach (Prunus persica L.) fruit ripening: aminoethoxyvinylglycine (AVG) and exogenous polyamines affect ethylene emission and flesh firmness. Physiol. Plant. 2022, 114, 472–481.
  2. Serrano, M., Martinez-Romero, D., Guillen, F., Valero, D., 2003. Effects of exogenous putrescine on improving shelf life of four plum cultivar. Postharvest Biol. Technol. 2003, 30, 259–271.”

Round 3

Reviewer 2 Report

The authors addressed all comments and the rebuttal provided were convincing. I have no more doubts about the paper.